# Innovative Strategies in Drug Repurposing to Tackle Intracellular Bacterial Pathogens

**DOI:** 10.3390/antibiotics13090834

**Published:** 2024-09-02

**Authors:** Blanca Lorente-Torres, Jesús Llano-Verdeja, Pablo Castañera, Helena Á. Ferrero, Sergio Fernández-Martínez, Farzaneh Javadimarand, Luis M. Mateos, Michal Letek, Álvaro Mourenza

**Affiliations:** 1Departamento de Biología Molecular, Área de Microbiología, Universidad de León, 24071 León, Spain; blort@unileon.es (B.L.-T.); jllav@unileon.es (J.L.-V.); pcase@unileon.es (P.C.); halvaf00@estudiantes.unileon.es (H.Á.F.); sfernm14@estudiantes.unileon.es (S.F.-M.); fjavad00@estudiantes.unileon.es (F.J.); luis.mateos@unileon.es (L.M.M.); 2Instituto de Biología Molecular, Genómica y Proteómica (INBIOMIC), Universidad de León, 24071 León, Spain; 3Instituto de Desarrollo Ganadero y Sanidad Animal (INDEGSAL), Universidad de León, 24071 León, Spain

**Keywords:** drug repurposing/drug repositioning, intracellular bacterial pathogens, host-directed therapy, antimicrobial resistance, high-throughput screening, multidrug resistance

## Abstract

Intracellular bacterial pathogens pose significant public health challenges due to their ability to evade immune defenses and conventional antibiotics. Drug repurposing has recently been explored as a strategy to discover new therapeutic uses for established drugs to combat these infections. Utilizing high-throughput screening, bioinformatics, and systems biology, several existing drugs have been identified with potential efficacy against intracellular bacteria. For instance, neuroleptic agents like thioridazine and antipsychotic drugs such as chlorpromazine have shown effectiveness against *Staphylococcus aureus* and *Listeria monocytogenes*. Furthermore, anticancer drugs including tamoxifen and imatinib have been repurposed to induce autophagy and inhibit bacterial growth within host cells. Statins and anti-inflammatory drugs have also demonstrated the ability to enhance host immune responses against *Mycobacterium tuberculosis*. The review highlights the complex mechanisms these pathogens use to resist conventional treatments, showcases successful examples of drug repurposing, and discusses the methodologies used to identify and validate these drugs. Overall, drug repurposing offers a promising approach for developing new treatments for bacterial infections, addressing the urgent need for effective antimicrobial therapies.

## 1. Introduction

The global burden of infectious diseases caused by intracellular pathogens remains a significant challenge to public health [1]. Pathogens such as *Listeria monocytogenes*, *Mycobacterium tuberculosis*, and *Staphylococcus aureus* have developed sophisticated mechanisms to survive and replicate within host cells [2,3], evading both the host immune response and traditional antimicrobial therapies. This evasion complicates treatment strategies and underscores the urgent need for innovative approaches to combat these infections [4].

Drug repurposing, the process of identifying new therapeutic uses for existing drugs, typically FDA-approved with established information on their toxicity, formulation, pharmacology, and potential side effects [5,6], has emerged as a promising strategy to address this challenge (Figure 1), especially pushed by the vast amount of information surrounding approved drugs [7]. In that sense, the emergence of COVID-19 has accelerated the application of drug repurposing strategies to find new and effective treatments for combating diseases [8,9]. Drug repurposing had primarily been accidental, lacking a systematic approach; however, the discovery of Viagra as an effective treatment for erectile dysfunction, instead of its original purpose as an antihypertensive drug, opened the door to a new and lucrative market, which was further propelled by the COVID-19 pandemic [10,11]. Repurposing offers a cost-effective and time-efficient alternative to the lengthy and expensive process of novel drug discovery [12] that reduces the investment needed for the development of new treatments and makes repurposing an attractive alternative for pharmaceutical companies [13]. Recent advances in high-throughput screening, bioinformatics, and systems biology have facilitated the identification of candidate drugs with potential efficacy against intracellular pathogens [14,15,16].

It is crucial to acknowledge that drug repurposing faces several significant challenges, with intellectual property barriers being foremost among them. From a profitability perspective, repurposed drugs often present difficulties, as generating profit can be challenging [10]. However, it is possible to secure patents and achieve profitability for a repurposed drug when a novel medical use is demonstrated. This requires successfully completing clinical trials to confirm the drug’s efficacy in its new application [10].

This review aims to provide a comprehensive overview of the current landscape of drug repurposing efforts targeting intracellular bacterial pathogens, which is an evolving field (Figure 2). We will discuss the mechanisms by which these pathogens evade conventional therapies, highlight key examples of successfully repurposed drugs, and explore the methodological approaches employed in identifying and validating these candidates.

## 2. *Staphylococcus aureus*

*Staphylococcus aureus* is a significant human pathogen responsible for a variety of infections. Approximately 30% of the human population is estimated to carry *S. aureus*, and it is the leading cause of bacteremia [17,18]. This pathogen expresses numerous virulence factors that aid in the colonization of eukaryotic cells. Attempts to develop vaccines targeting various *S. aureus* virulence factors have failed due to the bacterium’s ability to evade the immune response from multiple angles, rendering these vaccines ineffective [19]. Moreover, the misuse of antimicrobials has led to the emergence of multi-drug resistant bacteria [20]. Methicillin-resistant *S. aureus* (MRSA), in particular, was responsible for approximately 100,000 deaths related to antimicrobial resistance in 2019 [18]. The rapid evolution of bacteria poses significant challenges for developing new antimicrobials, making it often economically inefficient. Consequently, drug repurposing has emerged as a viable alternative [21]. This strategy has shown promise, particularly in the treatment of *S. aureus* infections.

*S. aureus* initially infects the skin and mucosal surfaces, and can eventually infect internal organs, causing pneumonia, endocarditis, osteomyelitis, and bacteremia [19]. More interestingly, *S. aureus* can act as both an extracellular and an intracellular pathogen, which significantly increases the difficulty of treating *S. aureus* infections. Once inside the cells, *S. aureus* can evade the phagosome and become free in the cytoplasm. Additionally, it controls the host adaptive immune response through a complex system of virulence factors expressed during infection [19]. For these reasons, *S. aureus* is a model for targeting host-directed therapies (HDT), as reactivating the immune system could be an effective strategy to kill intracellular bacteria.

A plethora of drugs target intracellular *S. aureus*, and interestingly, some have been extensively used for other purposes and are now repurposed to target intracellular infections (Table 1). Chlorpromazine (CPZ), originally used to treat psychosis, was the first drug discovered with anti-*S. aureus* activity among other intracellular pathogens [22]. It was demonstrated that the drug accumulates in macrophages, particularly in lysosomes, at higher concentrations than needed for its in vitro activity. The fusion between the phagosome (where the bacteria are usually located) and the lysosome triggers the antimicrobial activity of CPZ. However, the doses required for in vivo activity are beyond those clinically achievable, and CPZ becomes toxic when used chronically. Fortunately, another neuroleptic agent, thioridazine (TZ), has shown antimicrobial activity at clinically relevant doses with a similar mechanism of action as CPZ [22,23]. Thioridazine, a phenothiazine, exhibits activity in vitro against *S. aureus* and also acts against phagocytosed *S. aureus* by accumulating in macrophages [22,24].

Antimicrobials active against cell walls or cell wall synthesis are commonly used. Clomiphene, a fertility drug, antagonizes wall teichoic acid (WTA), showing activity against *S. aureus* and synergism with cell-wall-targeting antimicrobials [25].

Repurposing antimicrobial agents against new targets is another strategy to utilize existing compounds. Some agents are used alone or in combination with other antimicrobials to mitigate resistance. Interestingly, ebselen, a non-toxic seleno-organic drug with anti-inflammatory, antioxidant, and anti-atherosclerotic properties, has demonstrated activity against MRSA [26]. Additionally, while ebselen does not show activity against Gram-negative bacteria, it is effective against intracellular MRSA, highlighting its potential as a new anti-MRSA therapy [27].

The anthelmintic drug selamectin showed a Gram-positive antimicrobial profile with high activity against *S. aureus*, a low toxicity profile, and an 81.3% reduction in intracellular *S. aureus* loading, exhibiting a synergistic effect with ampicillin [28].

Celecoxib, an anti-inflammatory drug, inhibits cyclooxygenase-2 (COX-2) and also inhibits multidrug efflux pumps in *S. aureus*, increasing sensitivity to other antimicrobials. However, the antimicrobial mechanism of celecoxib remains unknown [29]. Diflunisal, another anti-inflammatory drug and a prostaglandin synthetase inhibitor, acts on the Agr quorum-sensing system, which plays a key role in pathological bone remodeling in *S. aureus* osteomyelitis [30]. Inactivation of the Agr system leads to significant bone destruction reduction in osteomyelitis patients [30].

Repurposing anticancer drugs as antimicrobials has several advantages, primarily due to their established safety spectrum and guaranteed intracellular activity. However, repurposing drugs from cancer to antibacterial therapies shares a common handicap: the appearance of resistance in both prokaryotic and eukaryotic cells [31]. Floxuridine, an inhibitor of riboside phosphorylase, and streptozotocin, a DNA synthesis inhibitor, were both repurposed to treat intracellular *S. aureus*. These compounds inhibit the SaeRS two-component system and demonstrated a significant survival rate in *S. aureus*-infected mice compared to the commonly used anthracycline antibiotic doxorubicin [32].

As an intracellular pathogen, *S. aureus* is exposed to reactive oxygen and nitrogen species (RONS), which usually reduce its viability [33]. Staphylotoxin, the golden carotenoid pigment of *S. aureus*, aids in ROS detoxification during infection. Interestingly, BPH-652, a cholesterol-lowering agent, binds to the dehydrosqualene synthase of *S. aureus*, reducing staphylotoxin and restoring the toxic effect of ROS against *S. aureus* [34]. Artemisinin, extensively used as an antimalarial drug, exhibits antibacterial activity via ROS generation; the addition of Cu^+^ ions enhances its activity through increased free radical production and DNA damage [35].

Ebselen, which shows non-selective activity against different cysteine-containing proteins, specifically inhibits thioredoxin reductase (TrxR) and thioredoxin (Trx) in *S. aureus*, resulting in toxic bactericidal activity against MRSA [36]. Similarly, auranofin decreases the reducing capacity of MRSA by inhibiting the TrxR system, making the bacteria more sensitive to oxidative stress [37]. Auranofin, an FDA-approved organogold drug, has proven effective in vivo against MRSA and intracellular *M. tuberculosis* at low micromolar concentrations [37]. It clears over 60% of intracellular *S. aureus*, demonstrating significantly higher efficacy compared to isoniazid and vancomycin, which only achieve a 30% clearance of intracellular bacteria [38].

Another strategy to fight intracellular pathogens is the induction of autophagy in infected cells, a treatment known as host-directed therapy [15,16]. Several compounds exhibit this activity. Ibrutinib, used for lymphocytic leukemia, decreases intracellular *Mycobacterium tuberculosis* and *S. aureus*, reducing MRSA viability by 90%. Ibrutinib inhibits the BTK/Akt/mTOR pathway during *M. tuberculosis* infections, triggering autophagy in macrophages [39]. Ibrutinib’s activity was further analyzed with other host-directed therapy compounds like dasatinib and crizotinib against *S. aureus*, acting as kinase inhibitors and blocking bacterial infection and intracellular proliferation [15]. Interestingly, crizotinib targets ATP production and presents broad-spectrum activity against Gram-positive bacteria, showing a synergistic effect with clindamycin and gentamicin [15,40].

Raloxifene, an estrogenic receptor agonist, has been repurposed against intracellular *S. aureus* due to its autophagy-inducing activity [41]. Raloxifene also prevents neutrophil cell death in response to phorbol 12-myristate 13-acetate (PMA), inhibiting PMA-induced ERK phosphorylation but not ROS production, which triggers the formation of neutrophil extracellular traps (NETs), known to contain antimicrobial substances aiding in intracellular pathogen clearance [41,42].

During infection, MRSA activates the MAPK/ERK pathway, which acts as a sensor for cell energy homeostasis and amino acid deprivation. Intriguingly, *S. aureus* appears capable of inducing autophagy to facilitate its own replication and proliferation [43]. Building on this hypothesis, dorsomorphin, an AMPK inhibitor, has shown activity against *S. aureus* as a host-directed therapy. It reduces autophagy and thereby decreases *S. aureus* intracellular survival [43]. Similarly, thapsigargin, an inhibitor of the SERCA protein (a Ca^2+^ pump in the endoplasmic reticulum), blocks autophagy, increases host cell viability, and reduces intracellular MRSA survival by two-fold [16].

**Table 1 antibiotics-13-00834-t001:** List of repurposed drugs against *S. aureus*: classification and mechanism of action.

Drug	Type	Mechanism of Action	Reference
Artemisinin	Antimalarial	Generates ROS and DNA damage	[35]
Auranofin	Anti-inflammatory	Inhibits thioredoxin reductase	[37]
BPH-652	Cholesterol-lowering	Binds dehydrosqualene synthase (CrtM)	[34]
Celecoxib	Anti-inflammatory	Inhibits multidrug efflux pumps	[29]
Chlorpromazine	Antipsychotic	Accumulates in lysosomes, triggering antimicrobial activity	[22]
Clomiphene	Fertility	Antagonizes wall teichoic acid	[25]
Crizotinib	Antitumoral	ATP production	[5,28]
Dasatinib	Antitumoral	Tyrosine kinase inhibitor	[15]
Diflunisal	Anti-inflammatory	Inhibits the Agr system	[30]
Dorsomorphin	AMPK inhibitor	Reduces autophagy and intracellular survival	[43]
Ebselen	Anti-inflammatory	Inhibits thioredoxin reductase and thioredoxin	[26]
Floxuridine	Antitumoral	Inhibits riboside phosphorylase	[32]
Ibrutinib	Antitumoral	Controls the MEK/ERK/c-JUN signaling pathway	[15]
Phenothiazine	Antipsychotic	Antagonism of dopamine D2 receptors	[11,13]
Raloxifene	Estrogen receptor agonist	Induces autophagy and inhibits neutrophil cell death	[41]
Selamectin	Anthelmintic	Shows high activity against *S. aureus* and *M. tuberculosis*	[28,41]
Streptozotocin	Antitumoral	DNA synthesis inhibitor	[28,32]
Thapsigargin	Ca^2+^ endoplasmic pump inhibitor	Increases host cell viability and reduces intracellular survival	[16,32]
Thioridazine	Antipsychotic	Accumulates in macrophages, triggering antimicrobial activity	[16,22,23]

## 3. *Mycobacterium tuberculosis*

*M. tuberculosis* is the bacterium responsible for tuberculosis (TB) and is considered the leading cause of death due to bacterial infections [14,44]. *M. tuberculosis* is adept at colonizing and surviving inside alveolar macrophages, modulating immune responses, and controlling macrophage maturation. Its metabolism is adapted to the challenging environment it encounters during infection. Recent reviews have focused on drug repurposing for *M. tuberculosis* treatment, specifically on drugs that can cross eukaryotic membranes to target intracellular *M. tuberculosis* [12,14,26]. One of the most prominent strategies includes host-directed therapies (Table 2) that have been extensively used against *M. tuberculosis* [12,14,45,46,47].

During infection, *M. tuberculosis* controls lysosome protease fusion proteins, downregulating them to aid colonization. Saquinavir, an anti-HIV drug, has been proposed as a repurposed drug against *M. tuberculosis* because it can restore cathepsin protease activity, including cathepsin S, in a dose-dependent manner in *M. tuberculosis*-infected macrophages [48]. Additionally, *M. tuberculosis* utilizes cholesterol, a crucial component of mammalian cell membranes, as an energy source essential for colonization. Cholesterol accumulation decreases membrane permeability to anti-tubercular drugs and blocks phagosome maturation by direct cholesterol uptake [49]. Consequently, statins, which treat hypercholesterolemia and atherosclerotic cardiovascular disease, have emerged as new anti-tubercular drugs. Atorvastatin, a 3-hydroxy-3-methyl-glutaryl-coenzume A (HMG-CoA) reductase inhibitor, enhances macrophage bactericidal effects [50]. Atorvastatin started a phase 2 clinical trial (ref. NCT06199921) this year. Simvastatin, another HMG-CoA inhibitor, reduces cholesterol uptake by *M. tuberculosis*, sensitizing the bacteria to other antimicrobial compounds and showing synergy with isoniazid and rifampicin [51].

Inhibiting efflux pumps is another strategy to enhance antimicrobial effects. *M. tuberculosis* detoxifies itself via a bacterial efflux pump-mediated process. Verapamil, an FDA-approved cardiovascular drug, has been successfully used as an adjuvant for TB treatment, acting as a mycobacterial efflux pump inhibitor and increasing the toxicity of isoniazid and rifampicin [52]; this drug was used in a recently finished Phase 1/2 clinical trial (ref. IRCT20170210032478N1). During *M. tuberculosis* infection, patients often suffer inflammatory responses, leading to tissue damage from neutrophil infiltration. Anti-inflammatory drugs, such as ibuprofen, have shown potential in reducing bacterial load in the lungs and improving survival in animal models, despite having no direct effect on *M. tuberculosis* [53]. Ibuprofen has progressed to two distinct Phase 2 clinical trials as an adjunctive treatment: one in combination with acetylsalicylic acid (currently ongoing, ref. NCT02781909), and another as a standalone treatment, which was completed in 2019 (ref. NCT03891901).

The activation of Toll-like receptors (TLR) is part of the innate defence against bacterial infections, but *M. tuberculosis* downregulates TLR activity. Vitamin D, which activates TLR, has shown anti-TB activity, highlighting the importance of sunlight in bacterial infections [54]. Repurposing anticancer drugs is another strategy against *M. tuberculosis*. Tamoxifen, an established anticancer drug, has been repurposed to induce autophagy in infected cells, promoting bacterial degradation in lysosomes [55]. Fluspirilene and pimozide, two diphenylbutylpiperidine-class antipsychotic drugs, elicit autophagy in infected cells, promoting bacterial clearance. Pimozide also affects ROS generation and inhibits STAT5, reducing the presence of cytokine-inducible SH2-containing protein on infected phagosomes [56].

*M. tuberculosis* uses ABL and other related host tyrosine kinases for entry and intracellular survival. Imatinib, a tyrosine kinase inhibitor used in cancer therapy, has been successfully tested against *M. tuberculosis* in vitro and in vivo [57] and it was in a Phase 2 clinical trial testing its safety, pharmacokinetics and hematologic effect alone or in combination with isoniazid and rifabutin, finishing the study in 2022 (ref. NCT03891901). The development of kinase inhibitors has been pursued for years due to their importance in intracellular pathogen colonization. Kuijl et al. (2007) have developed several kinase inhibitors with good antimicrobial profiles against *M. tuberculosis* and *Salmonella enterica* serovar Typhimurium [58].

Promoting autophagy in infected cells is a classic method to combat intracellular pathogens. Bazedoxifene, a selective estrogen receptor modulator (SERM) used in cancer therapy, has been repurposed against *M. tuberculosis*, reducing the intracellular growth of the bacteria in THP-1 macrophages [59]. Vitamin D also shows antimicrobial activity via autophagy induction by activating the human cathelicidin microbial peptide (CAMP) [60]. Vitamin D3 induces antimicrobial activity through autophagy and nitric oxide production, acting through cathelicidin and TLRs, which induce a nitric oxide burst [54,61]. The good activity shown against *M. tuberculosis* opened the door to three different clinical trials (refs. NCT00507000 and NCT01130311); one of them was finalized in 2023 (ref. NCT01992263).

Nitroimidazofurans, originally used as radiosensitizers in cancer chemotherapy, are bicyclic molecules that show anti-TB activity in the nanomolar range, similar to isoniazid [62]. PA-824 has demonstrated activity in murine models treated orally with the compound, thus PA-824 was used in a Phase 2 clinical trial that finalized in 2022 (ref. NCT02256696). Chronic inflammation, often resulting from infection, aids *M. tuberculosis* infection [63]. Reducing deleterious inflammation can make current antibiotherapies more effective. Metformin, a biguanide used for type 2 diabetes, has emerged as a new HDT therapy by activating a major energy-sensing kinase and reducing inflammation by limiting the proliferation of inflammatory cells [63]. Metformin was in a Phase 1/2 clinical trial that finalized in 2023 (ref. NCT05215990). Metformin, in combination with statins, has shown synergistic effects [64].

Once *M. tuberculosis* colonizes pulmonary tissues, it often forms granulomas characterized by hypoxia and necrosis, similar to solid tumors. This induces the production of vascular endothelial growth factor (VEGF), promoting new vessel formation and aiding TB progression. Bevacizumab, a humanized monoclonal antibody used to neutralize VEGF in some cancers, has been used to reduce TB progression and disease lesions in animal models [65].

Reintroducing “old” antimicrobials is an effective strategy against multidrug-resistant bacteria due to their known pharmacokinetics and pharmacodynamics. Sulphonamides have been studied as new TB therapies. Sulfadiazine, an analogue of para-aminobenzoic acid, acts as a dihydropteroate synthase inhibitor, blocking the folic acid synthesis pathway in *M. tuberculosis* and showing high in vitro activity, suggesting its potential as a second-line antibiotic in combination with other treatments [66].

Targeting cell wall synthesis is a common strategy against bacterial infections. Arabinosyl transferase (EmbC) is crucial for lipoarabinomannan biosynthesis, a major polysaccharide in the cell envelope. Ethambutol, which targets EmbC, is part of TB treatment but often encounters resistance. Terlipressin, originally a vasoactive drug for low blood pressure, has been identified as a new anti-TB drug targeting EmbC in a different region than ethambutol, showing high binding affinity and reduced resistance potential [67]. In silico analysis has identified 29 compounds with anti-TB activity, with fluvastatin showing intracellular activity at moderate concentrations, highlighting the power of bioinformatics in antimicrobial discovery [68].

Anthelmintic drugs like avermectins, used against helminths, insects, and arachnids, have shown activity against mycobacteria. Ivermectin, selamectin, and moxidectin have demonstrated in vitro activity against MDR *M. tuberculosis*, showing high specificity and oral activity, making them promising antimicrobial therapies [69]. Nitazoxanide, an FDA-approved anthelmintic and antiparasitic drug, inhibits intracellular *M. tuberculosis* growth by modulating host cell immune responses [70] and it was in a Phase 2 clinical trial to study its bactericidal activity (ref. NCT02684240).

Phenothiazine-derived antipsychotic drugs have shown high antitubercular activity with minimal side effects, showing no cytotoxicity in vitro and in vivo [71]. Carbapenems, such as biapenem, which was involved in a Phase 1 clinical trial (ref. NCT01702649), and tebipenem, have shown activity against *M. tuberculosis* by targeting L,D-transpeptidases, [72]. Faropenem has shown even better transpeptidase affinity and good in vivo antimicrobial activity [72], results that opened the door to clinical trials; thus, Faropenem was used in a Phase 3 clinical trial (ref. NCT01937832). Carbapenems show synergistic effects with rifampin, with doripenem, biapenem, and rifampin combinations showing the best results [73].

Artemisinin and its derivative artesunate, known for their anti-malarial properties, also show anti-TB activity with confirmed low toxicity and good pharmacokinetic profiles in murine models [74]. Imidazopyridine amides, such as compound Q203, target the cytochrome bc1 complex and have shown effective inhibition of *M. tuberculosis* in vitro and in mouse models [75]. Moreover, Q203 was utilized in a Phase 2 clinical trial finished in 2019 that evaluated its bactericidal activity, safety, tolerability, and pharmacokinetics following multiple oral doses (ref. NCT03563599).

*M. tuberculosis* experiences oxidative stress inside macrophages, making its redox mechanisms, including NADH dehydrogenase type II and thioredoxin reductase (TrxR), critical. Chlorpromazine, an NADH inhibitor, and gatifloxacin, a fluoroquinolone, that was in a Phase 3 clinical trial (ref NCT00216385), have shown effective synergistic antimicrobial effects, indicating their potential as dual therapies [76]. Auranofin, a TrxR inhibitor, has shown significant efficacy against both intracellularly replicating and non-replicating *M. tuberculosis* [37]. In fact, Auranofin was used in a Phase 2 clinical trial (ref. NCT02968927) that finalized in 2020.

**Table 2 antibiotics-13-00834-t002:** List of repurposed drugs against *M. tuberculosis*: classification and mechanism of action.

Drug	Type	Mechanism of Action	Reference
Artemisinin	Antimalarial	Generates ROS and DNA damage	[74]
Artesunate	Antimalarial	Unknown	[74]
Atorvastatin	Statin	Enhances macrophage bactericidal effects	[50]
Auranofin	Anti-inflammatory	Inhibits thioredoxin reductase	[37]
Bazedoxifene	Estrogen receptor modulator	Reduces intracellular growth	[59]
Bevacizumab	Antitumoral	Neutralizes VEGF	[65]
Biapenem	Antimicrobial	Targets cell wall synthesis	[72]
Chlorpromazine	Antipsychotic	NADH dehydrogenase type II inhibitor	[76]
Faropenem	Antimicrobial	Targets cell wall synthesis	[72]
Fluspirilene	Antipsychotic	Elicits autophagy	[56]
Fluvastatin	Statin	Targets intracellular activity	[68]
Gatifloxacin	Antimicrobial	DNA gyrase	[76]
Ibuprofen	Anti-inflammatory	Reduces inflammatory response	[53]
Ibrutinib	Antitumoral	Inhibits the BTK/Akt/mTOR pathway, triggering autophagy	[39]
Imatinib	Antitumoral	Inhibits tyrosine kinases	[57]
Imidazopyridine amides (Q203)	Antimicrobial	Cytochrome bc1 complex	[61]
Ivermectin	Anthelmintic	Inhibits intracellular growth	[69]
Metformin	Antidiabetic	Activates energy-sensing kinase and reduces inflammation	[63]
Moxidectin	Anthelmintic	Unknown mechanism of action	[55]
Nitroimidazofuran (PA-824)	Antimicrobial	Bacterial nitroreduction	[49]
Nitazoxanide	Anthelmintic	Modulates host immune responses	[70]
Phenothiazine	Antipsychotic	NADH inhibitor	[57,62]
Pimozide	Antipsychotic	Affects ROS generation and inhibits STAT5	[56]
Saquinavir	Antiviral	Restores cathepsin protease activity	[48]
Selamectin	Anthelmintic	Unknown mechanism of action	[55]
Simvastatin	Statin	Reduces cholesterol uptake, showing synergy with antimicrobials	[51]
Sulfadiazine	Antimicrobial	Inhibits folic acid synthesis	[66]
Tamoxifen	Antitumoral	Induces autophagy	[55]
Tebipenem	Antimicrobial	Targets cell wall synthesis	[72]
Terlipressin	Vasoactive	Targets EmbC for cell wall synthesis	[67]
Verapamil	Cardiovascular	Inhibits bacterial efflux pump	[39]
Vitamin D	Vitamin	Activates TLR and induces autophagy	[54]

## 4. *Listeria monocytogenes*

*Listeria monocytogenes* (LMO) is a Gram-positive, saprophytic bacterium that can exist both intracellularly and extracellularly, categorizing it as a facultative intracellular pathogen. It is the causative agent of listeriosis, a serious infection typically leading to meningitis, septicemia, and neonatal death [77]. Upon infection, LMO penetrates the intestinal epithelial barrier and migrates to the lamina propria [78]. LMO enters host cells via endocytosis; once inside the phagosome, it escapes into the cytoplasm where it can reside [79]. LMO infection triggers an inflammatory response, apoptosis, necrosis, pyroptosis, and autophagy in the infected cells [78]. As a host response, both major histocompatibility complex (MHC) I and II molecules produce immune responses involving CD4 and CD8 T cells and the production of interferon γ (IFN γ).

Given the high fatality rate of LMO among food-borne pathogens (13%) and a 97.1% hospitalization rate [80], repurposing drugs to combat LMO infections could be a cost-effective strategy. Similar to other intracellular pathogens like *S. aureus* and *M. tuberculosis*, LMO can control infection by modulating the host-cell immune response, making host-directed therapies (HDT) potentially effective alone or in combination with antimicrobials for new treatments.

Neurological compounds have shown promise against LMO (Table 3). In a screening of 68 neurological compounds, 26 exhibited inhibitory effects on LMO internalization in eukaryotic cells. Notably, thioridazine (an antipsychotic) and bepridil (a calcium channel inhibitor) inhibited LMO’s escape from the vacuole, resulting in bacterial death [81]. Pimozide, previously reported, blocks LMO entry into host cells, phagosome escape, and cell-to-cell spread, indicating its potential as an effective treatment by targeting multiple stages of infection [82]. However, no in vivo results for pimozide have been reported.

Repurposing antimicrobials is another strategy against LMO (Table 3). Griseofulvin, an antifungal drug, has demonstrated activity against LMO, although it lacks activity against *S. aureus* or *Bacillus subtilis*. This discovery has led to modifications of its structure to increase the activity of various derivatives [83]. These compounds preferentially target GyrB, DNA topoisomerase IV, and thymidylate kinase, showing low or negligible cytotoxicity [83]. Despite these efforts, no in vivo studies have been conducted on these compounds, and their stability remains untested.

## 5. *Salmonella enterica* Serovar Typhimurium

*Salmonella* Typhimurium is a Gram-negative bacterium that causes gastroenteritis and severe systemic infections in humans and other animals. It infects both epithelial cells and macrophages. Epithelial cell infection is primarily driven by the *Salmonella* Pathogenicity Island 1 (SPI-1), whereas infection of macrophages and neutrophils occurs through phagocytosis, leading to the formation of a modified phagosome known as the *Salmonella*-containing vacuole (SCV) [84]. Within host cells, *S.* Typhimurium releases various effectors to aid its intracellular survival [85]. Additionally, *S.* Typhimurium infection alters host cholesterol biosynthesis [86].

The use of host-directed therapies (HDT) is widely applied for *Salmonella* due to its role as a model for host-pathogen interactions [86] (Table 4). Statins, which affect host cholesterol metabolism, have been shown to negatively impact bacterial growth. For instance, lovastatin blocked intracellular proliferation of *S.* Typhimurium in macrophages and murine models [86].

Anti-inflammatory drugs are another powerful tool for repurposing against *S.* Typhimurium as they modulate the host-cell response. Diclofenac sodium, an anti-inflammatory drug that inhibits prostaglandin G/H synthase 1 and 2, exhibited antimicrobial activity against *S.* Typhimurium and other bacteria both in vitro and in vivo. This compound, alone or in combination with other drugs, showed significant antimicrobial activity against several pathogens, including *Shigella* spp. and *Salmonella* spp. Diclofenac sodium demonstrated a synergistic effect with streptomycin and trifluoperazine, reducing mortality in mice from 100% without treatment to less than 10% in treated groups [87].

High-throughput screening (HTS) of various compounds has been a promising method for identifying effective repurposed drugs against *S.* Typhimurium. Doxapram, amoxapine, and trifluoperazine were identified using this technique. Trifluoperazine, a dopamine D2 receptor blocker used as an antipsychotic, accumulates in macrophages and reduces *S.* Typhimurium intracellular survival by potentially targeting the host autophagy pathway [88]. In the same line, loperamide, an antidiarrheal drug, blocked the intracellular replication of *S.* Typhimurium not by directly affecting the bacteria, but by inducing autophagy in *S.* Typhimurium-infected cells [89].

Bioinformatic tools are also utilized to identify inhibitors of specific bacterial proteins or virulence factors. For instance, Joshi et al. (2022) analyzed over 1900 compounds as potential *S.* Typhimurium dihydrofolate reductase inhibitors. From this, eight compounds were further analyzed, and four (duvelisib, amenamevir, lifitegrast, and nilotinib) were identified as the most promising inhibitors [90].

HTS of different antimicrobials against intracellular *S.* Typhimurium yielded interesting results. Since antimicrobials can have different effects in vitro versus during infection, a new technique was developed to evaluate the activity of 1600 antimicrobials against intracellular *S. enterica*. Nucleoside analogs, including doxifluridine, fluorouracil, azacitidine, and carmofur, were among the most effective. Additionally, bromperidol, metergoline, ciclopirox and ethopropazine showed significant activity at the lowest concentration of 64 µg/mL [84]. Metergoline, the most promising agent, disrupted the cytoplasmic membrane potential decreasing ATP levels. This alkaloid, produced by fungi, was effective in murine models, particularly in the spleen, liver, cecum, and colon [84].

**Table 4 antibiotics-13-00834-t004:** List of repurposed drugs against *S. enterica*: classification and mechanism of action.

Drug	Type	Mechanism of Action	Reference
Amoxapine	Antidepressant	Increase the levels of norepinephrine and serotonin	[74]
Amenamevir	Antiviral	Inhibits helicase-primase complex	[75]
Azacitidine	Antitumoral	Disrupts cytoplasmic membrane potential	[84]
Bromperidol	Antipsychotic	Dopamine D2 receptor antagonist	[70]
Carmofur	Antitumoral	Inhibitor dihydropyrimidine dehydrogenase	[84]
Ciclopirox	Antifungal	Inhibits iron-dependent enzymes	[70]
Diclofenac sodium	Anti-inflammatory	Inhibits prostaglandin G/H synthase	[74]
Doxapram	Respiratory stimulant	Stimulates respiratory chemoreceptors	[74]
Doxifluridine	Antitumoral	Inhibits thymidylate synthase	[70]
Duvelisib	Antitumoral	Inhibits δ and γ isoforms PI3K	[75]
Ethopropazine	Anticholinergic	Blocks muscarinic acetylcholine receptors	[70]
Fluorouracil	Antitumoral	Disrupts cytoplasmic membrane potential	[84]
Lifitegrast	Anti-inflammatory	Integrin agonist	[75]
Loperamide	Antidiarrheal	Promotes autophagy	[89]
Lovastatin	Anti-cholesterol	Inhibits HMG-CoA reductase	[72]
Metergoline	Antipsychotic	Serotonin receptor antagonist	[70]
Nilotinib	Antitumoral	Inhibits tyrosine kinase	[75]
Trifluoperazine	Antipsychotic	Accumulates in macrophages, targets autophagy pathway	[88]

## 6. Other Intracellular Pathogens

Intracellular pathogens have unique strategies for surviving and proliferating within host cells. For instance, *Chlamydia* forms inclusions; *Legionella* and *Coxiella* occupy lysosomes; and *Shigella* and *Rickettsia* inhabit the cytosol [84].

*Shigella flexneri*, a common gastrointestinal pathogen, has been targeted for drug repurposing. As an intracellular pathogen, *S. flexneri* is susceptible to drugs that induce autophagy in host cells. Capsaicin, a herbal compound known to induce autophagy, has shown effectiveness in reducing *S. flexneri* intracellular growth both in vitro and in vivo, with a good safety profile in murine models [91]. Diclofenac, an anti-inflammatory drug, has also demonstrated activity against *S. flexneri* in murine models, prompting further research that advanced to clinical trials. These efforts culminated in a Phase 4 clinical trial, which revealed diclofenac’s efficacy in women with urinary tract infections (UTIs) who were treated for 30 days. The trial results suggest that diclofenac may have potential as a therapeutic option for UTI management, warranting further exploration and consideration in clinical practice [92].

In efforts to combat *Yersinia pestis*, the causative agent of plague, a library of 780 FDA-approved compounds was screened for their ability to block intracellular colonization. Among the most effective compounds were doxapram, amoxapine, and trifluoperazine, which reduced intracellular survival in vitro and increased animal survivability in vivo, despite not having direct antimicrobial activity [88].

The concept of broad-spectrum host-directed therapies (HDT) is particularly relevant for intracellular pathogens. Broad-spectrum HDT drugs can assist host cells in mounting an immune response against various intracellular bacteria (Table 5). However, it is crucial to first test the toxicity of these compounds, as they often alter host cell behavior and can be toxic.

Czyz et al. identified several repurposed FDA-approved drugs with broad-spectrum activity against *Legionella pneumophila*, *Brucella abortus*, *Rickettsia conorii*, and *Coxiella burnetii*. Notably, drugs related to GPCR signaling, kinases, calcium inhibitors, and sterol/hormones emerged as potential intracellular antimicrobial agents. The effectiveness of these drugs varied depending on the pathogen, highlighting the importance of understanding each bacterium’s infection mechanism. For example, blocking cholesterol trafficking was effective against *C. burnetti* and *L. pneumophila* but not against *B. abortus* and *R. conorii* [93]. Statins, used for lowering cholesterol levels, have shown effectiveness against various pathogens, as previously mentioned. Simvastatin, in particular, was also effective against *Chlamydia pneumoniae*, a human respiratory pathogen. By decreasing host cell isoprenoid and cholesterol levels, simvastatin disrupted the intracellular trafficking of cholesterol, reducing the intracellular survival of *C. pneumoniae*. This mechanism also applied to *Salmonella enterica*, which lacks a cholesterol synthesis pathway [94].

Trifluoperazine, initially identified as a repurposed drug against *Yersinia pestis*, did not exhibit direct antimicrobial activity but showed the best in vivo activity. It was hypothesized that its mechanism of action could be effective against other pathogens. Indeed, trifluoperazine was also tested in vivo against *Salmonella* Typhimurium and *Clostridium difficile* increasing animal survivability in both cases [88].

**Table 5 antibiotics-13-00834-t005:** List of repurposed drugs against several intracellular pathogens: classification, mechanism of action, and target pathogens.

Drug	Type	Mechanism of Action	Pathogen	Reference
Capsaicin	Herbal compound	Induces autophagy	*S. flexneri*	[91]
Diclofenac sodium	Anti-inflammatory	Inhibits prostaglandin G/H synthase	*Shigella* sp.	[87]
Doxapram	Respiratory	Blocks intracellular colonization	*Y. pestis*	[88]
Simvastatin	Statin	HMG-CoA inhibitor	*C. pneumoniae*	[94]
Trifluoperazine	Antipsychotic	Accumulates in macrophages, targets autophagy pathway	*C. pneumoniae*, *C. difficile, Y. pestis*	[88]

## 7. Discussion

The rapid evolution of bacteria and the increasing prevalence of antimicrobial resistance pose significant challenges to the development of new antimicrobials. This situation has led many pharmaceutical companies to deprioritize further research in this area, increasing the risk of new pandemics emerging worldwide [18,20]. Drug repurposing has emerged as a viable and cost-effective strategy to address these challenges by using existing drugs with known safety profiles (Figure 1). This review has explored various repurposed treatments against a range of intracellular pathogens, including *S. aureus*, *M. tuberculosis*, *L. monocytogenes*, *S. enterica*, *S. flexneri*, *Y. pestis*, *C. pneumoniae*, *C. difficile* and *Shigella* sp., focusing on drugs initially developed for cancer, anti-inflammation, autophagy modulators, ROS related drugs, and antimicrobials that have been repurposed against other bacteria. The limited number of repurposed drugs currently in clinical trials, with only two ongoing studies and one successfully completed suggests that prior approval for other uses does not necessarily expedite their authorization for new indications [8]. This could be attributed to various factors, such as the lack of efficacy in human trials or the possibility that these trials have progressed beyond the clinical stage and are advancing to the next phase. Another challenge may lie in the fact that the bactericidal mechanisms of action for many of these drugs are still under investigation [11]. Additionally, the continued effectiveness of existing antibiotic therapies for most patients, the limited understanding of potential resistance that microorganisms might develop against repurposed drugs, and the lack of sufficient funding could also contribute to the slow progress. Despite these challenges, drug repurposing remains a faster and more cost-effective approach to drug research and development [13].

This review focuses on repurposed drugs that are active against intracellular pathogens. These microorganisms share a common trait: they can alter cellular responses to evade bacterial clearance once inside the host cells, often by modifying autophagy or inflammatory responses. As a result, some of the drugs discussed here are host-directed therapies (HDTs) because they do not directly target the bacteria but instead modulate the immune response of the eukaryotic cell. This strategy has shown promising results due to its broad-spectrum effects, including cross-species activity and effectiveness against multidrug-resistant pathogens. While there is no definitive evidence yet on how host-directed therapy (HDT) will reduce antimicrobial resistance, many experts support the idea that targeting eukaryotic cells may make the development of resistance less likely. Since the mechanisms of resistance in pathogens primarily involve prokaryotic processes, the likelihood of resistance emerging against HDT is considered to be lower [4,14,15,45]. Consequently, some of the drugs under study (such as the antipsychotics pimozide, phenothiazine, thioridazine, trifluoperazine, and chlorpromazine; the anti-inflammatory drugs auranofin and diclofenac; the classic antimalarial drug artemisinin; the anticancer drug ibrutinib; and the anthelmintic selamectin) are active against various microorganisms. Their broad-spectrum activity is linked to their mechanisms of action. For example, some anticancer and antipsychotic drugs induce autophagy in infected cells, leading to bacterial clearance, although the exact mechanisms by which these drugs induce autophagy remain unclear [28,39,69,74].

The mechanisms of action of anti-inflammatory drugs are more diverse, often involving the inhibition of bacterial efflux pumps, which can enhance the effectiveness of conventional antibiotic therapies. Additionally, the anti-inflammatory effects of these drugs benefit patients by reducing systemic inflammation, which can otherwise be fatal. Finally, intracellular pathogens are exposed to oxidative bursts that can damage proteins and kill bacteria, making bacterial redox mechanisms crucial for their survival. Several of the drugs discussed, such as auranofin and ebselen, inhibit essential bacterial redox mechanisms such as TrxR, thereby reducing intracellular survival [26,37].

Drug repurposing is an active field of research that is growing exponentially in the number of published papers and, consequently, in the number of new repurposed drug candidates. Despite these efforts, several gaps remain to be addressed, such as the emergence of bacterial resistance or tolerance, potential side effects in patients with other underlying conditions, and the need for optimization of dosing and formulation. Additionally, the challenge of securing adequate funding and navigating the complex regulatory pathways for approval further complicates the development and widespread adoption of repurposed drugs. Addressing these challenges will be critical to fully realizing the potential of drug repurposing as a viable and efficient approach to treating a wide range of diseases [5,11,13].

Overall, drug repurposing offers a promising avenue for developing new treatments against bacterial infections. By leveraging the existing pharmacological knowledge and safety profiles of these drugs, we can accelerate the discovery of effective therapies and address the urgent need for new antimicrobials in the fight against bacterial pathogens.

## Figures and Tables

**Figure 1 antibiotics-13-00834-f001:**
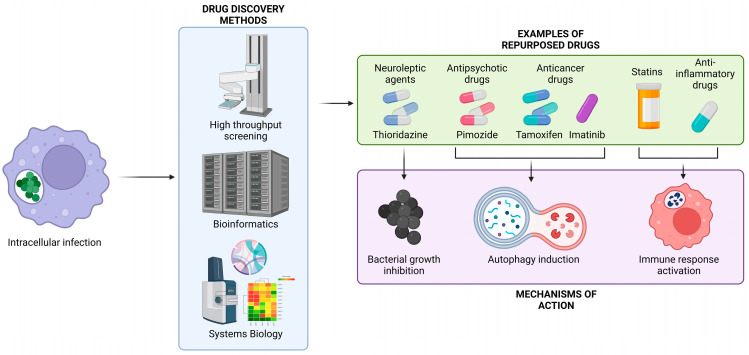
Schematic representation of the most common process to repurpose drugs against intracellular pathogens, as well as the most common mechanisms of action.

**Figure 2 antibiotics-13-00834-f002:**
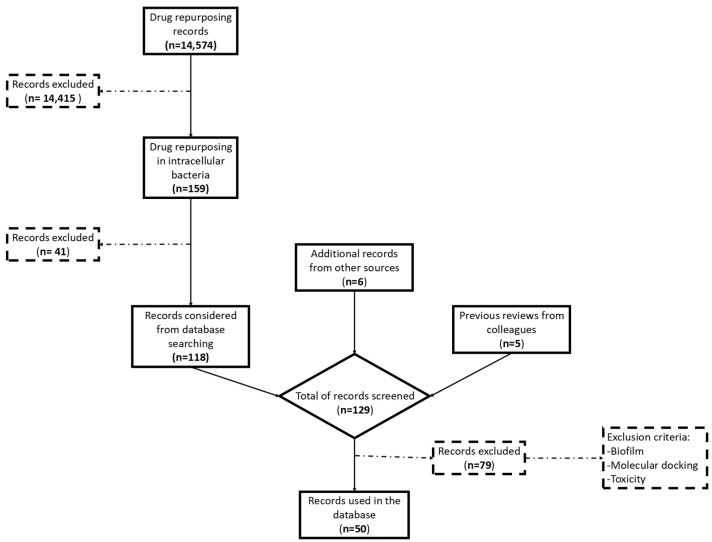
Summary of the scientific results obtained from different sources (PubMed, conferences and colleagues). The search was narrowed to drugs repurposed against bacteria and subsequently to intracellular bacteria. Only about 1% of the articles on drug repurposing focused on intracellular pathogens. Exclusion criteria were based on the off-target effects, toxicity, or lack of tested activity.

**Table 3 antibiotics-13-00834-t003:** List of repurposed drugs against *L. monocytogenes*: classification and mechanism of action.

Drug	Type	Mechanism of Action	Reference
Bepridil	Antitumoral	Inhibits calcium channel inhibitor	[67]
Griseofulvin	Antifungal	Disruptor of microtubules	[69]
Pimozide	Antipsychotic	Dopamine D2 receptor antagonist	[68]
Thioridazine	Antipsychotic	Dopamine D2 receptor antagonist	[67]

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
