# Peer review of "Innovative Strategies in Drug Repurposing to Tackle Intracellular Bacterial Pathogens"

_antibiotics, 2024, doi:10.3390/antibiotics13090834_

Round 1

Reviewer 1 Report

Comments and Suggestions for Authors

The review article entitled "Drug Repurposing: Innovative Strategies Against Intracellular Bacterial Pathogens” provides a compelling case for the Drug repurposing concept as antibiotics which has been recently explored as a strategy in the discovery of a novel drugs and medications.

The utilizing of a high-throughput screening, and bioinformatics have been identified with potential efficacy against intracellular bacteria. In this review the authors were highlighting the complex mechanisms of these pathogens use to resist conventional treatments, showcases successful examples of drug repurposing, and they well discussed the methodologies which used to identify and validate these drugs.

Overall, this work is a valuable resource for researchers in the field and contributes meaningfully to the ongoing efforts to combat antibiotic resistance to be antibiotics journal after justifying the following minor points:

·       The authors are encouraged to updates their title to be more attractive for readers and researchers as the following suggestions:

1-     Innovative Strategies in Drug Repurposing to Tackle Intracellular Bacterial Pathogens

2-     Reimagining Antibiotic Therapy: Repurposed Drugs Against Intracellular Bacteria

·       The authors should increase the numbers of the references since this is an review article work 

·       They should add the used database and years of collection for the used references accordingly

·       The number of keywords should be reduced as well as you should foucs on the main idea of this review, so keywords like antitumoral agents and antipsychotic drugs better to be removed   

·       In the whole manuscript the authors should use the italic style for In vitro in vivo and bacterial strains

·       In the first and second paragraphs of the introduction it is recommended to increase the number of the references as well as don’t use 5 references for the same paragraph like 2-6 references you should separate them accordingly as possible

·       Figure 1 has very important summarized data, could you present these data in better style, the arrows should be thicker as well as the borders of each box

·       The authors should explain well the concept of the mentioned repurposed drugs were applied and reach any clinical trials or approved form the FDA or this is just an research article data ???

·       The discussion section should be improved well

·       Usually, we don’t use a figure in the conclusion section

Best wishes

Author Response

We thank the reviewer for the time dedicated to improve our manuscript. We have carefully reviewed and revised the manuscript in response to the reviewer’s comments. Additionally, we have made further adjustments to other areas of the text that we identified for improvement. Please refer to the revised manuscript and the comments below for more details.

Comment 1: The review article entitled "Drug Repurposing: Innovative Strategies Against Intracellular Bacterial Pathogens” provides a compelling case for the Drug repurposing concept as antibiotics which has been recently explored as a strategy in the discovery of a novel drugs and medications.

The utilizing of a high-throughput screening, and bioinformatics have been identified with potential efficacy against intracellular bacteria. In this review the authors were highlighting the complex mechanisms of these pathogens use to resist conventional treatments, showcases successful examples of drug repurposing, and they well discussed the methodologies which used to identify and validate these drugs.

 Response 1: We thank the reviewer for the positive comments and especially the time to read and propose valuable comments to increase the overall quality of the manuscript.

Comment 2: Overall, this work is a valuable resource for researchers in the field and contributes meaningfully to the ongoing efforts to combat antibiotic resistance to be antibiotics journal after justifying the following minor points:

 Response 2: We thank the reviewer for the positive comment, we have addressed the points the reviewer highlighted.

  • Comment 3: The authors are encouraged to updates their title to be more attractive for readers and researchers as the following suggestions:

1-     Innovative Strategies in Drug Repurposing to Tackle Intracellular Bacterial Pathogens

2-     Reimagining Antibiotic Therapy: Repurposed Drugs Against Intracellular Bacteria

      Response 3: We agree with the reviewer, and we have changed the title to “Innovative Strategies in Drug Repurposing to Tackle Intracellular Bacterial Pathogens”

  • Comment 4: The authors should increase the numbers of the references since this is an review article work 

      Response 4: We thank the reviewer for the comment. We have refined our search as described in Figure 2 (new version), focusing on repurposed drugs that have demonstrated efficacy and low or undetected toxicity against intracellular pathogens. Additionally, we have incorporated several more articles, particularly in the introduction and throughout the text. The number of references has increased from 79 to 94, which we believe now includes all the most recent and relevant research in the field.

  • Comment 5: They should add the used database and years of collection for the used references accordingly

      Response 5: We thank the reviewer for their comment. In addition to using the PubMed database, we have incorporated knowledge gained from colleagues during conferences, meetings, and direct discussions, as summarized in the new Figure 2 (this information and the new references have also been added to the figure’s footnotes and image). However, since this is not a systematic review, we believe that including this information in the main text may not be relevant.

      Comment 6: The number of keywords should be reduced as well as you should foucs on the main idea of this review, so keywords like antitumoral agents and antipsychotic drugs better to be removed   

      Response 6: We agree with the reviewer, and the keynotes have been reduced and refined (lines 31 to 33)

  • Comment 7: In the whole manuscript the authors should use the italic style for In vitro in vivo and bacterial strains

      Response 7: We apologize for the mistakes that we introduced in the text, we have changed all the necessary words to italics as suggested by the reviewer.

  • Comment 8: In the first and second paragraphs of the introduction it is recommended to increase the number of the references as well as don’t use 5 references for the same paragraph like 2-6 references you should separate them accordingly as possible

      Response 8: We agree with the reviewer. Therefore, we have added more references in the introduction and expanded the section (Lines 45-54; 57-58 and 67-72). Additionally, we have updated references 2-6, placing them in the specific parts of the paragraph where they are most relevant.

  • Comment 9: Figure 1 has very important summarized data, could you present these data in better style, the arrows should be thicker as well as the borders of each box

      Figure 9: We thank the reviewer for the valuable comment, we have changed the Figure 1 (now figure 2) following the reviewer´s instructions.

  • Comment 10: The authors should explain well the concept of the mentioned repurposed drugs were applied and reach any clinical trials or approved form the FDA or this is just an research article data???

      Response 10: We have added some information regarding the clinical trials related with the drugs included in the manuscript. Unfortunately, mostly of them are still in basic research stages so they dind´t go to clinical trials. However, there are some drugs that have been or are in clinical trials against M. tuberculosis and S. flexneri, we have added this information in the specific sections.

      Modifications added are in Lines: 245-246; 254-255; 259-262; 275-277; 289-291; 295-296; 300-301; 330-331; 334; 337-338; 345-347; 344-346; 350-351; 352-355; 450-456, of the revised manuscript.

  • Comment 11: The discussion section should be improved well

      Response 11: We thank the reviewer for the comment. In response, we have expanded the discussion section, adding information on the strengths and weaknesses of drug repurposing, as well as general details on the mechanisms of action of the various drug types and microorganisms reviewed. Additionally, we have included information on some gaps we identified in the literature (Lines 493-495; 502-542; 559-568 of the revised manuscript).

  • Comment 12: Usually, we don’t use a figure in the conclusion section

      Response 12: We agree with the reviewer, and we have changed the figure 2 to introduction section line 62 of the revised manuscript.

Reviewer 2 Report

Comments and Suggestions for Authors

Author Response

We thank the reviewer for the time dedicated to improve our manuscript. We have carefully reviewed and revised the manuscript in response to the reviewer’s comments. Additionally, we have made further adjustments to other areas of the text that we identified for improvement. Please refer to the revised manuscript and the comments below for more details.

Comment 1: The increasing worldwide prevalence of infectious illnesses, especially those brought on by bacterial infections inside cells, highlights the critical need for novel treatment strategies. Due to their ability to elude both conventional medicines and human immune responses, these bacteria present serious threats to public health. As a result, drug repurposing has become a viable tactic, investigating the possibilities of already-approved, well-studied medications for novel therapeutic uses.

This manuscript explores the possibility of repurposing drugs to fight intracellular bacterial infections, with a particular emphasis on many important pathogens, including Salmonella enterica serovar Typhimurium, Mycobacterium TB, Listeria monocytogenes, and Staphylococcus aureus. The study focuses on the effectiveness of several therapeutic classes in blocking these pathogens, such as statins, neuroleptic medicines, anti-cancer therapies, and anti-inflammatory drugs.

Finding antibacterial qualities in currently available medications is one of the manuscript's main themes. By examining the mechanisms of action behind these repurposed medicines, the authors provide insight into their potential to target survival strategies employed by bacteria. For example, neuroleptic medications such as chlorpromazine and thioridazine show effectiveness against Staphylococcus aureus by obstructing the intracellular trafficking of germs. Tamoxifen and imatinib, two anti-cancer medications, have antibacterial qualities because they promote autophagy, a cellular mechanism that breaks down intracellular microorganisms.

Response 1: We thank the reviewer for the good summary of the review and we agree with the importance of the drug repurposing as a new strategy to combat bacterial infections.

Comment 2: Overall, the work fits within the journal's focus, offers a thorough overview of medication repurposing in the setting of intracellular bacterial infections, and merits publishing in an antibiotics journal. However, before publication, several areas need more investigation. It is important include information about other recently published reviews that are connected to the area, the breadth and limits of previous reviews in comparison to the present article, and how this particular review fills a gap in the field. Furthermore, direct comparisons between the effectiveness of repurposed medications and approved therapies are a useful contribution. To ensure completeness, it is also necessary to address the possibility of drug resistance developing, difficulties that are a major worry in antimicrobial therapy, and the inclusion of clinical data where it is available to support therapeutic claims of repurposed medications.

Response 2: We thank the reviewer for the positive comments. In response to the comments, we have made extensive revisions to the manuscript to address all the points raised. Additionally, we have incorporated newly released publications in the field of drug repurposing. The number of references has increased from 79 to 94, with most of the new references added to the introduction section to better establish the state of the art in this field.

Comment 3: Strengths:

               o The manuscript addresses a critical and timely issue in infectious disease             treatment.

               o It provides a comprehensive overview of drug repurposing potential.

               o Well-chosen examples illustrate the concept effectively.

Response 3: We thank the reviewer for highlighting the strengths of our review.

Comment 4: Areas of improvement

  1. Clinical Relevance: The efficacy of these repurposed medications in practical situations is not sufficiently supported by clinical research. The text would be strengthened with more convincing clinical evidence or data from ongoing trials.

  1. Comparative Efficacy: The effectiveness of repurposed medications and existing

standard-of-care therapies is not sufficiently contrasted in the study. This comparison is

essential to assessing their possible benefits and drawbacks.

  1. Resistance Development: There is insufficient discussion of the possibility of drug

resistance developing against these medications. In light of antibiotic resistance, this is a

serious omission.

Response 4: We also thank the reviewer for pointing out the weaknesses of the manuscript, which has helped us improve its quality before publication. We believe that the revised version addresses the previous weaknesses. We have added information about clinical trials and the clinical relevance of repurposed drugs. Additionally, we have highlighted the comparison between current antibiotic therapies and new repurposed drug therapies in the introduction and conclusion sections. We have also extensively reviewed the importance of resistance development. Please see below for detailed information about the sections and information added.

Comment 5: The English language is appropriate and understandable

Response 5: Thanks

Comment 6: Title and Abstract: The content is appropriately and clearly reflected in the title. The abstract offers a succinct and thorough summary. The abstract offers a thorough summary, highlighting the possibilities for therapeutic repurposing as well as the threats to public health posed by intracellular bacterial infections. The efficiency of several medications against intracellular bacteria is mentioned, and the approaches used to find and validate these medications are covered. The topic of the study is succinctly summarized in the well-written abstract.

Response 6: We thank the reviewer for the comment. However, we have changed the title following reviewer 1 advice to Innovative Strategies in Drug Repurposing to Tackle Intracellular Bacterial Pathogens”

Comment 7: Introduction: It would be beneficial to include freshly released relevant reviews and research. They are recommended in the reference section below. The introduction lays forth the background information, explains why the study is necessary, and expresses the goals in unambiguous terms. More information on the many sources and exclusion criteria used to obtain scientific results is beneficial. The introduction addresses the critical need for novel therapies as well as the worldwide burden of infectious illnesses brought on by intracellular microorganisms. Because of the proven safety profiles, pharmacokinetics, and manufacturing procedures of repurposed pharmaceuticals, it proposes drug repurposing as a potential method. With a focus on particular pathogens like Listeria monocytogenes, Mycobacterium tuberculosis, and Staphylococcus aureus, the introduction effectively establishes the paper's strong background. Still, a few pertinent recent reviews should be included.

Response 7: In agreement with the reviewer comment, we have included the four references provided by the reviewer (Refs. 7, 9, 10, and 89 in the revised manuscript), as we believe they are relevant to our review. Additionally, we have added other references that enhance the clarity of the state of the art in the field. The revised manuscript now contains 94 references, up from 79.

Comment 8: Content: The work offers a thorough analysis of the literature and is rationally and well structured. The ideas raised are backed up by pertinent studies.

Response 8: Thanks for the kind comment.

Comment 9: Methodology: The technique, being a review study, entails a thorough examination of the body of current literature. To guarantee repeatability and openness, however, the authors should clearly identify the databases they examined, their inclusion and exclusion criteria, and their search approach.

Response 9: We thank the reviewer for their comment. Since this is not a systematic review, we decided not to include a detailed methodology for our search process. However, to provide clarity, we have added the figure 1 (Figure 2 in the revised manuscript) that outlines our search methodology and the criteria used for excluding papers. In addition to using the PubMed database, we have incorporated knowledge gained from colleagues during conferences, meetings, and direct discussions, as summarized in the new Figure 2 (this information has also been added to the figure’s footnotes).

Comment 10: Conclusions: The authors stress the need for more research to hasten the development of efficient medicines and conclude that medication repurposing offers a potential strategy for creating novel therapeutics for bacterial infections. The authors should note any gaps in the literature and offer guidance on the anticipated future course of medication repurposing in order to further the area.

Response 10: We thank the reviewer for their comment. We agree with the feedback and, following the advice of the other two reviewers, we have extensively revised the discussion section. We have added specific information about the gaps found in the literature (Lines 559-568 of the revised manuscript). Additionally, we have included details on the lack of efficacy of current antibiotic therapies and the challenges associated with antibiotic research (Lines 493-495).

Comment 11: References: Related recently published reviews/studies are missing and their inclusion would help.

Response 11: Following the reviewer comment we have completed the reference section with 15 new references that have filled some gaps of the previous version of the manuscript.

Comment 12: 1. Line 56:

o References needed for summary of scientific results obtained from different sources.

Response 12: We thank the reviewer for their comment. However, we believe there may have been a misunderstanding. The sentence in question pertains to the sources of information used to find papers, such as PubMed and insights from colleagues.

Comment 13: 2. Line 73:

o Reference(s) needed to clarify several studies

Response 13: We have removed that sentence in the revised version (lines 125-126) as we found it irrelevant.

Comment 14: 3. Line 143-44:

o Support the statement and claim with suitable reference(s)

Response 14: We thank the reviewer for the comment, we have added references 15 and 16 in line 196 of the revised version to support our statement.

Comment 15: 4. Lines 173:

o States recent reviews that indicate a need of multiple reference citation, but only ref. 4 is mentioned.

Response 15: We thank the reviewer for the comment. We have added ref 26 and 12 of the revised version in line 227 of the revised manuscript.

Comment 16: 5. Line 227-228:

o Support the statement with suitable reference(s)

Response 16: We thank the reviewer for the comment, we have added reference 61 in line 289 to support our statement.

Comment 17: 6. Table 2:

o For Moxidectin and Selamectin, a key mechanism of action enhancing its high

activity will help readers and for uniformity. If it s unknown, then stating it would

help.

Response 17: we thank the reviewer for the comment, we have changed both mechanisms of action to unknow in the revised table 2 of the manuscript.

Comment 18: 7. Line 350-356:

o Support the claim with suitable reference(s)

Response 18: We thank the reviewer for the comment. We have completed the sentence with ref 83 in line 430 of the revised version.

Comment 19: 8. Line 475, 478, 484, 521, 556, 567, 569, 577, 587, 596, 605, 614, 619, 626, 629, :

o Doi: address and page numbers in some are missing in reference. Please check

and fix for consistency.

Response 19: We thank the reviewer for pointing out these mistakes, we have corrected when possible since some of the references are online only they have not page numbers.

Reviewer 3 Report

Comments and Suggestions for Authors

Thanks for the opportunity to review this manuscript. As a reviewer, In this review article author discusses several examples of existing drugs that have been repurposed to combat intracellular bacterial pathogens. The review highlights the potential of drug repurposing as a cost-effective and time-efficient approach to developing new treatments for intracellular bacterial infections. The review covers several examples of existing drugs that have been repurposed to combat intracellular S. aureus infections, such as chlorpromazine, thioridazine, clomiphene, etc. This demonstrates the potential of drug repurposing in this field.

Here are some comments:

 1.     In this review article author does not provide a comprehensive overview of the underlying mechanisms of action for these repurposed drugs in discussion. More detailed information related to the specific molecular targets and pathways would be beneficial to understand the potential activities of these drugs.  

2. This review primarily focuses on S. aureus, but it lacks discussion on the potential of drug repurposing against other intracellular bacterial pathogens, such as M. tuberculosis. Expanding the scope to include other clinically relevant intracellular bacteria would provide a more comprehensive understanding of the field.

3. Here the author does not address the potential challenges and limitations associated with drug repurposing, such as the need for further optimization, potential off-target effects, and the need for clinical validation. Discussing these aspects would help readers better understand the practical considerations and future research directions in this field.

Comments on the Quality of English Language

 Minor editing of English language required.

Author Response

We thank the reviewer for the time dedicated to improve our manuscript. We have carefully reviewed and revised the manuscript in response to the reviewer’s comments. Additionally, we have made further adjustments to other areas of the text that we identified for improvement. Please refer to the revised manuscript and the comments below for more details.

Comment 1: Thanks for the opportunity to review this manuscript. As a reviewer, In this review article author discusses several examples of existing drugs that have been repurposed to combat intracellular bacterial pathogens. The review highlights the potential of drug repurposing as a cost-effective and time-efficient approach to developing new treatments for intracellular bacterial infections. The review covers several examples of existing drugs that have been repurposed to combat intracellular S. aureus infections, such as chlorpromazine, thioridazine, clomiphene, etc. This demonstrates the potential of drug repurposing in this field.

Response 1: We thank the reviewer for reading the manuscript and for the time dedicated to helping us improve its overall quality. However, we respectfully disagree with the assessment that the review is focused solely on S. aureus. Our review encompasses information on various intracellular pathogens, with S. aureus and M. tuberculosis being two of the most studied targets for drug repurposing. While the review does give considerable attention to these two pathogens, it also covers other relevant pathogens such as L. monocytogenes, S. enterica, S. flexneri, Y. pestis, C. pneumoniae, C. difficile, and Shigella spp.

 Comment 2: In this review article author does not provide a comprehensive overview of the underlying mechanisms of action for these repurposed drugs in discussion. More detailed information related to the specific molecular targets and pathways would be beneficial to understand the potential activities of these drugs. 

Response 2: We appreciate the reviewer’s comment and agree that understanding the mechanisms of action is crucial for comprehending the role of repurposed drugs. However, for many of the cases presented, detailed information about the mechanisms of action is not yet available, as these drugs are still under investigation. We have revised the conclusions to include information about the mechanisms of action for some of the drugs, although we were unable to provide details for all of them. The updated conclusion section can be found between lines 493-495; 502-542; 559-568 of the revised manuscript.

Comment 3: This review primarily focuses on S. aureus, but it lacks discussion on the potential of drug repurposing against other intracellular bacterial pathogens, such as M. tuberculosis. Expanding the scope to include other clinically relevant intracellular bacteria would provide a more comprehensive understanding of the field.

Response 3: We thank the reviewer for the comment. However, we respectfully disagree with the reviewer's perspective. Our review addresses a range of intracellular pathogens, with a focus on S. aureus and M. tuberculosis due to the extensive volume of work available for these pathogens. While we have included less information on other intracellular pathogens, this is because the body of research on them is significantly smaller. In the revised manuscript, we have updated the discussion to include information on all the pathogens reviewed in the main text (Lines 499-500). We hope the reviewer understands our approach and finds the new version satisfactory.

Comment 4:  Here the author does not address the potential challenges and limitations associated with drug repurposing, such as the need for further optimization, potential off-target effects, and the need for clinical validation. Discussing these aspects would help readers better understand the practical considerations and future research directions in this field.

Response 4: In agreement with reviewer´s comment we have included information about the limitations of drug repurposing in both the introduction (Lines 67-72) and the discussion section (Lines 502-514; 559-568) of the revised manuscript.

Round 2

Reviewer 1 Report

Comments and Suggestions for Authors

the authors were improved the manuscript very well according to our comment and suggestions